# Treatment Effect Performance of the X-Learner in the Presence of Confounding and Non-Linearity

Bevan I. Smith [1,*], Charles Chimedza [2] and Jacoba H. Bührmann [1,*]

[1] The School of Mechanical, Industrial and Aeronautical Engineering, University of the Witwatersrand, Johannesburg 2000, South Africa
[2] School of Statistics and Actuarial Science, University of the Witwatersrand, Johannesburg 2000, South Africa
[*] Correspondence: bevan.smith@wits.ac.za (B.I.S.); joke.buhrmann@wits.ac.za (J.H.B.)

**Abstract:** This study critically evaluates a recent machine learning method called the X-Learner, that aims to estimate treatment effects by predicting counterfactual quantities. It uses information from the treated group to predict counterfactuals for the control group and vice versa. The problem is that studies have either only been applied to real world data without knowing the ground truth treatment effects, or have not been compared with the traditional regression methods for estimating treatment effects. This study therefore critically evaluates this method by simulating various scenarios that include observed confounding and non-linearity in the data. Although the regression X-Learner performs just as well as the traditional regression model, the other base learners performed worse. Additionally, when non-linearity was introduced into the data, the results of the X-Learner became inaccurate.

**Keywords:** treatment effects; counterfactuals; confounding

## 1. Introduction

In order to improve graduation rates, universities employ various interventions for students to participate in such as extra face-to-face tutorials [1], supplementary videos [2], orientation programs [3], early alert processes [3], and grants [3]. The aim is to improve various student outcomes, such as retention and graduation [3] or final grades [1,2,4]. The challenge however, is being able to accurately estimate if the interventions improve outcomes, i.e., to estimate true treatment effects. Interventions in higher education settings, such as extra tutorials or online videos, are generally observational studies, not randomized studies [3–7]. This is due to students self-selecting to attend the intervention and not being randomly assigned. The problem with observational studies is that the estimation of the treatment effects generally introduces bias thereby not yielding accurate treatment effect estimations [8]. Traditional ways of estimating treatment effects in observational studies include regression, propensity score matching (PSM) [9,10], and inverse probability of treatment weighting (IPTW) [10]. Regression methods fit a regression model to the observed data, including confounding features, and estimate the treatment effect coefficient. Matching methods aim to measure treatment effects by creating a counterfactual group by matching observations in treated and control groups. However, this causes the dataset to shrink due to needing to match cases that balance the treated and control groups.

Recently, a new method called the X-Learner has been introduced, that uses machine learning to estimate treatment effects in observational studies [11]. It has been applied in higher education studies to estimate treatment effects [1,4,12]. This method estimates treatment effects not by balancing the control and treated datasets to obtain counterfactuals (as in PSM), but by predicting counterfactuals and including them in the treatment effect estimation. Because of not needing to balance the groups, the full dataset is preserved and no observations are lost [13]. Although this counterfactual prediction method (X-Learner) shows promise, the problems are the following:

- The above-mentioned studies [1,4,12], were carried out on real-life data where the true treatment effects were not known. Studies are therefore required where the true treatment effect is known, in order to evaluate the accuracy of the X-Learner method. Smith et al. [1] attempted to validate the X-Learner method by comparing it with PSM estimations; however the ground truth average treatment effect (ATE) was still not known. Beemer et al. [4] and Beemer et al. [12] applied this method to student performance data but did not attempt to validate with alternative methods.
- Kunzel et al. [11] did carry out simulation studies with known treatment effects; however they did not compare these results with traditional regression methods for estimating treatment effects. This is important because as we will see, the X-Learner method is a multi-step complicated computation, and if it does not provide clear benefits over traditional methods, then it might not be worthwhile.

Therefore, the main aim of this study was to carry out a critical study of the X-Learner method by performing simulation studies where the true treatment effect, $ATE_{true}$, was known and where the X-Learner was compared with traditional regression methods. This study is important for determining to what extent this method can be used to measure treatment effects in higher education (and elsewhere) on real life datasets. Due to observational studies generally possessing bias due to confounding, all simulations in this study included a single feature confounding. We also aimed to study how non-linearity affected the results.

The X-Learner is known as a meta-learner, since it can use any supervised learning model as a base learner, to perform computations [13]. The models used in our study were linear regression, lasso regression, and random forest. This is, as far as we know, the first study to simulate this method investigating how confounding, non-linearity and different supervised learning models affect the treatment effect estimations. Kunzel et al. [11] simulated unobserved confounding whereas we simulated observed confounding. They used random forest and Bayesian regression trees (BART), whereas we introduced linear regression and lasso regression. The results of the X-Learner were compared with the ground truth treatment effects, as well as standard methods for estimating treatment effects, namely traditional regression and a naive method.

*Main Contributions*

The X-Learner was compared with traditional regression methods for estimating treatment effects. The X-Learner used three base learners, namely linear model, regularized linear model (lasso), and random forest to estimate known treatment effects. In all simulations, the traditional regression methods outperformed every X-Learner in terms of estimating the true treatment effect. In general, the X-Learner linear model performed better than the lasso model which performed better than the random forest base learner, when the simulated data were linear. When the data were non-linear, random forest tended to perform better than lasso. The main contribution of this work is that for the simulated condition of single feature confounding and non-linearity, the X-Learner presents no clear benefits over traditional regression methods. Although the X-Learner using a regression base learner, performs just as well as the traditional regression model, it requires a number of extra steps to perform just as well. The other base learners performed much worse than the traditional methods. Additionally, when non-linearity was introduced into the data, the results of the X-Learner became inaccurate. It may perform well under different circumstance, but did not outperform traditional methods under the range of scenarios presented in this study.

## 2. Related Work

Estimating treatment effects in studies in education were traditionally carried out using approaches, such as regression [5,14,15] and propensity score matching [1,16]. The use of machine learning methods for predicting student performance has been seen for approximately twenty years in the literature [17,18]. Other online studies using machine

learning included [19–21]. However, the use of machine learning for predicting treatment (or causal) effects in educational studies is relatively new [2,4].

As introduced above, estimating counterfactuals are the key to causal inference. Counterfactuals can be used to investigate causality in questions like "If there had been more police patrols would the attack at the bus stop have been prevented?" Regression type models have been used to evaluate counterfactuals [22]. The use of counterfactuals in machine learning has become popular of late, and is drawing a lot of attention[23].

Recently, new machine learning methods called meta-learners have been introduced to the literature where machine learning methods have been applied to estimating treatment effects. These meta-learners are methods that use machine learning models, such as random forest [2,24], regression trees [11,25], k-nearest neighbors [11] as base learners to estimate treatment effects. Currently, there are three primary meta-algorithms in the literature, namely the S-Learner, T-Learner, and X-Learner [11,13].

The use of these meta-learners has not seen widespread use in studies in education. Until now, only three studies have been found that use the X-Learner in educational settings [1,4,12]. Beemer et al. [4] applied the X-Learner to real life data with the hope of estimating individual treatment effects for students. The aim was to carry out personalized learning. In a follow up study, Beemer et al. [12] used the same methods to compare treatment effects of online vs. in-person teaching. Smith et al. [2] applied the X-Learner to predict treatment effects for students but went a step further by aiming to validate the results by comparing with the more traditional propensity score method. The problem with the above three studies is that the X-Learner was applied to real life data without knowing the true treatment effect. This is the challenge when applying causal inference to real life data: the ground truth treatment effects are not known. This forms the main impetus for the need of this study. How does the X-Learner perform when we do know the true treatment effect? This is done via simulation.

The fundamental idea behind meta-learners is to train machine learning models on treated and control groups separately and then use these models in various combinations to estimate treatment effects. In the literature there have been similar ideas that train separate models on the control and treated groups. For example, Jennifer Hill trained Bayesian additive regression trees (BART) on linear and non-linear data in simulation studies, with the aim of estimating treatment effects [25]. Athey and Imbens trained separate random forest models on the treated and control groups [26]. Both of these methods estimated treatment effects using Equation (1). The difference between their method and the X-Learner method, is the estimation of the counterfactuals as shown above. Furthermore, both studies assume unconfoundedness whereas ours introduces observed confounding.

## 3. Observational Studies and Treatment Effects

Traditional ways of estimating treatment (causal) effects are either to carry out a randomized control trial (RCT), the gold standard of experimental studies [8], or controlling for observed confounders in observational (non-randomized) studies [27]. Although the benefits of randomized trials are evident, we often only have access to observational data, which generally produces biased treatment effects due to self-selection confounding [8,28]. An example of an observational study is where we are aiming to estimate the treatment effect of online educational videos on the grades of university students. The treated group would be those students that watched supplementary online videos and the control group would be those that did not. Therefore, the treatment is the watching of videos. The important idea here is that in an observational study, the students self-selected to watch the videos thereby potentially introducing confounding.

Treatment effects can be measured via the average treatment effect, *ATE* seen in Equation (1), which is the difference between the expected outcome of the treated and control groups:

$$ATE = E(Y_t) - E(Y_c). \tag{1}$$

However, in observational studies, *ATE* is generally not the true treatment effect, but a naive effect containing bias due to confounding. Equation (1) will therefore, in general, not estimate the true treatment effect. The true *ATE* in observational studies can be computed using Equation (2) [28],

$$ATE = \pi(E(Y_t) - E(\hat{Y}_t)) + (1 - \pi)(E(\hat{Y}_c) - E(Y_c)), \tag{2}$$

where $\hat{Y}_t$ and $\hat{Y}_c$ are the counterfactual outcomes (introduced earlier) for the treated and control groups, respectively, and $Y_t$ and $Y_c$ are the actual outcomes of the treated and control groups, respectively. The quantity $\pi$ refers to the fraction of observations receiving the treatment, called the participation rate. It is important to note that the counterfactual outcomes cannot be measured (this is the fundamental problem of causal inference [29]), and the challenge here is to find a method that can predict the counterfactuals.

## 4. Confounding Bias in Observational Studies

In observational studies, bias due to confounding occurs when both the outcome $Y$ and the treatment $T$ are caused by a common parent $X$ [13], shown in the directed acyclic graph (DAG) in Figure 1. An example of confounding is where students are assigned (or self-select) an intervention ($T$), say extra videos or tutorials, based on their current grades ($X$), which also affect the outcome ($Y$). The problem is that $T$ possesses inherent bias that does not result in the true ATE when applying Equation (1). The phenomenon seen in Figure 1 is also known as the backdoor path, where association from $T$ to $Y$ is not only direct but via $X$ (the backdoor).

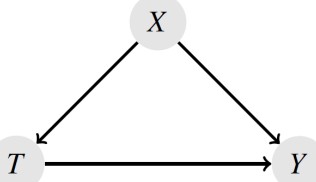

**Figure 1.** Directed acyclic graph showing confounding.

Ideally, we want $T$ to be independent of $X$ so that the unbiased treatment effect [8] of $T$ on $Y$ can be measured, shown in Figure 2. The scenario in Figure 2 can take place if we randomize which students take the treatment and thereby eliminate any causal relationship $T$ has with $X$. This is however not always possible due to ethical reasons: we can not and should not refuse any students access to an academic intervention.

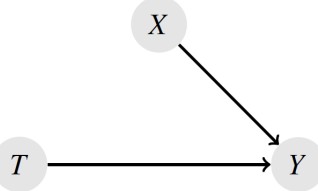

**Figure 2.** Directed acyclic graph showing no confounding.

If we cannot randomize, as in the case of observational studies, we can block the backdoor path between $T$ and $Y$, either by regression or matching techniques [13].

*The Counterfactual*

The idea behind either randomization in randomized trials or controlling/adjusting (from observational studies), is that we are trying to create a counterfactual scenario. A counterfactual is where everything remains the same, and only one thing changes, i.e., the treatment. In confounding in Figure 1 this was not the case. As $T$ was changing, so was $X$:

*T* was a function of *X*. The goal is that only *T* would change. That is where Figure 2 comes in which is to keep everything else fixed, while only changing the treatment. Practically, we create counterfactuals by having two groups that are statistically identical, and then only change the treatment. These groups are known as treatment and control.

The whole aim of the X-Learner therefore is to try to compute counterfactuals in order to estimate true treatment effects. The method is presented next.

## 5. X-Learner Method

For a dataset where a treatment was administered, the method for predicting the counterfactuals and computing the treatment effect is as follows:

1. Split the entire dataset into treated and control groups.
2. Train a machine learning model $M_t$ on the treated group and a model $M_c$ on the control group.
3. Predict treated group counterfactual, $\hat{Y}_t$, by feeding the treated group input features, $X_t$, into the control group model $M_c$. Predict control group counterfactual, $\hat{Y}_c$, by feeding the control group input features, $X_c$, into the treatment model $M_t$.
4. Estimate the true treatment effect using Equation (2).

This method therefore uses supervised learning to estimate counterfactuals by predicting what the treated group would have obtained, had they received the control and what the control group would have obtained, had they received the treatment. The idea behind this method is that we feed the input features of a certain group, into a model that was trained on other features of another group. That model then "imprints" its characteristics onto the input features. Therefore, for example, feeding the control group input features into the treated model, will imprint the treatment group characteristics onto the control group to thereby estimate the control group counterfactual. Similar for the treatment group. Furthermore, as mentioned earlier, using this method with Equation (2) preserves the use of the entire dataset [13].

## 6. Simulations

To evaluate the X-Learner, we performed various simulations. All simulations included a single observed confounding feature and half the simulations were based on linear relationships and half were non-linear. The aim was to ascertain how well the X-Learner performs when confounding and non-linearity are present, with different base machine learning models. The models being used as base learners were linear regression, lasso regression, and random forest.

### 6.1. Dataset Features

Our generated dataset, shown in Table 1, comprised seven input features (including a treatment indicator), and a target output feature. The dataset mimics data from a university course, where the input features represent grades and demographic features of students, and the output is a final course grade. The binary variables `Gender` and `Bursary` were simulated using a Binomial distribution since the number of observation in each groups can be varied [30]. A normal distribution was assumed for continuous data. This is consistent with other studies where normality was assumed in the data generating process [2,31]. Table 1 presents the input features and how the data for each feature was generated. The three grades features, $x_1$, $x_3$, and $x_6$ describe grades for three courses (A, B, and C) that a student could have in a semester, with mean values of 50%, 45%, and 70%, respectively. The Age feature describes the age distribution of first year students. Gender is slightly skewed to one side with a probability of 0.6, as in the case of real-world data [1]. We further included a government Bursary feature with a probability of receiving the bursary equal to 0.3. The sample size generated for each simulation was 1000 and for each simulation we carried out 500 iterations. Other educational studies used samples sizes between approximately 200 and 10,000 [32,33] and, therefore, our sample size of 1000 was considered suitable [34,35].

Furthermore, typical cohort sizes of first year engineering courses are approximately 1000 [2].

**Table 1.** Description of generated data.

| Name | Description | Distribution |
|---|---|---|
| $x_1$ | Grades A | Normal, $\mu = 50$, sd = 5 |
| $x_2$ | Age | Normal, $\mu = 20$, sd = 2 (minimum of 18) |
| $x_3$ | Grades B | Normal, $\mu = 45$, sd = 6 |
| $x_4$ | Gender | Binomial, prob = 0.6 |
| $x_5$ | Bursary | Binomial, prob = 0.3 |
| $x_6$ | Grades C | Normal, $\mu = 70$, sd = 7 |
| $T$ | Treatment | see Equation (4) |

The following simulations were run which are elaborated on in the subsequent sections.

1. Simulation A: Confounding; linear dataset.
2. Simulation B: Confounding, non-linear (squared) dataset.

For each category above, we tested the methods on a range of true treatment effects ($ATE_{true}$) and participation rates, $\pi$. The following simulations were run which are elaborated on in the subsequent sections.

### 6.1.1. Simulation A: Linear and Confounding

We first generated a linear dataset with single feature confounding (as per Figure 2). The structural causal model (SCM) is shown in Equation (3). $Y$ was generated from the input features plus a Gaussian error of zero mean and standard deviation of 1. All variable were simulated to be independent and no multi-collinearity existed between features. The only variables that had dependent relationships were treatment variables $T$, $x_3$, and the outcome $Y$.

$$Y = \beta_0 + \beta_1 x_1 + \beta_2 x_2 + \beta_3 x_3 + \beta_4 x_4 + \beta_5 x_5 + \beta_6 x_6 + ATE_{true}T + \mathcal{N}(0,1) \qquad (3)$$

Arbitrary values for the coefficients were chosen as follows: $\beta_0 = 0.4$, $\beta_1 = 0.6$, $\beta_2 = 0.4$, $\beta_3 = 0.9$, $\beta_4 = 0.7$, $\beta_5 = 0.4$, and $\beta_6 = 0.4$. These coefficients were used for all subsequent simulations.

Confounding was introduced via a single feature, $x_3$, to simulate an observational study. That is, we made $x_3$ a parent of both $T$ and $Y$, as shown in Figure 1, introducing a backdoor path. Treatment was now dependent on a single observable feature. Confounding was simulated as per the treatment assignment in Equation (4): if any student obtained below 41% for their $x_3$ grade, they would attend the intervention, i.e., receive treatment and be encoded as 1. If any student obtains above 49% they would not attend the intervention, and be encoded as 0. Between 41% and 49% we generated a probability of 0.5 based on a normal distribution, indicating that in this region there is a 50% probability of the students receiving the treatment.

$$T = \begin{cases} 1 & \text{if } x_3 < 41\% \\ 0 & \text{if } x_3 > 49\% \\ 0.5 \ \ probability & \text{between 41\% and 49\%} \end{cases} \qquad (4)$$

### 6.1.2. Simulation B: Non-Linear (Squared) and Confounding

The previous simulation was linear in all the features. We next generated non-linearity by squaring $x_1$ in the data generating process. Since we are not fitting a model to existing data but generating non-linear data, we chose to simply square $x_1$ as shown below. The remainder of the features all had a linear relationship with $Y$ and coefficients as per above.

Confounding was introduced in the same manner as the linear simulations. The SCM is seen in Equation (5):

$$Y = \beta_0 + \beta_1 x_1^2 + \beta_2 x_2 + \beta_3 x_3 + \beta_4 x_4 + \beta_5 x_5 + \beta_6 x_6 + ATE_{true}T + \mathcal{N}(0,1) \tag{5}$$

6.1.3. Range of $ATE_{true}$ Values

A further aim was to study a range of true ATE values. $ATE_{true}$ was varied as follows: [50, 10, 1]. The aim of varying $ATE_{true}$ as shown was to determine if large treatment effects (50) would affect the method differently to smaller ones (10, 1).

After generating the data, we then estimated *ATE* using the Equation (6) which is similar to Equation (2):

$$ATE_{sim} = \pi(E(Y_t) - E(\hat{Y}_t)) + (1 - \pi)(E(\hat{Y}_c) - E(Y_c)) \tag{6}$$

We therefore computed $ATE_{sim}$ using the X-Learner method (see pseudocode in Section 6.1.4) for a range of true treatment effects.

6.1.4. Pseudocode for Simulating the X-Learner Method

The pseudocode for the simulations is presented below (Pseudocode 1).

---

Pseudocode 1 for each run of a simulation of the X-Learner method

---

1. Generate data: $x_1$ to $x_6$, *T* and *Y* as per Table 1 and Equations (3) and (5) depending on type of simulation.
2. Split dataset into treated and control groups.
3. Train the three models (linear model $M_t$, lasso $L_t$, random forest $R_t$) on treated group; train the three models (linear model $M_c$, lasso $L_c$, random forest $R_c$) on control group.
4. Feed treated group input features $x_1$ to $x_6$ into linear model $M_t$, lasso $L_t$, random forest $R_t$ to predict treated group counterfactual $\hat{Y}_t$ for each model;
   feed control group $x_1$ to $x_6$ into $M_t$, $L_t$ and $R_t$ to predict control group counterfactual $\hat{Y}_c$ for each model.
5. Compute $ATE_{sim}$ for the three models, as per Equation (2) and store.

---

## 7. Linear Regression, Lasso and Random Forest Models

This study made use of three models to carry out the X-Learner predictions in the simulations: linear regression, lasso regression, and random forest. Linear regression was used as a base model. Lasso regression and random forest were used to see if they perform better than linear regression when non-linearity and confounding are present in the data.

The linear regression model is a basic parametric model using ordinary least squares to estimate the coefficients.

The lasso model [36] is a regularized regression model that is able to shrink to zero correlated features by introducing a penalty on the coefficients as follows:

$$\lambda \sum_{j=1}^{p} |\beta_j| \tag{7}$$

where $\lambda$ is a penalty, *p* is the number of features, and $\beta$ refers to the coefficients in the parametric model. When training the lasso models, 10-fold cross validation was employed to obtain the $\lambda$ values that minimize the mean-squared-error during training. Once the $\lambda$ values were obtained, they were utilized in the lasso models to carry out predictions.

The third model random forest, is an ensemble method that generates multiple decision trees and averages the outcome for regression problems such as the one in this study. It is a non-parametric machine learning model that generates each tree by carrying out bootstrap aggregating on the observations and random selection of a subset of the features. This allows for the trees in the ensemble to be decorrelated and produce superior results when

compared with single decision trees. They also are able to handle non-linearity well due to their non-parametric nature [36].

## 8. Baseline Methods

The X-Learner used three base models, linear (LM), lasso, and random forest (RF). We further compared the X-Learner (using the three models) against two baseline methods: one being the traditional regression method (Reg) and the other being the naive method (Naive). The regression method is simply to fit a regression model to the data, including the confounding variable and to read off the treatment indicator coefficient. The naive method computes the average treatment effect as per Equation (1), by simply computing the average outcome of treatment and control and calculating the difference.

Please note that this linear regression baseline method should not be confused with using regression inside the X-Learner, as per Section 7. They baseline regression method is to fit a linear regression model to the entire dataset as is done traditionally.

There are therefore five methods for estimating *ATE* and comparing it with the true *ATE*: three X-Learner methods and two baseline methods.

## 9. Software

The simulations were run in R [37] using the RStudio integrated development environment [38]. The glmnet [39] and caret [40] packages were used for lasso and random forest, respectively.

## 10. Results

Here we report two main results. The first is to compare the *ATE* estimation of the different methods using Equation (6). As mentioned, we simulated this 500 times and plotted the distribution of $ATE_{sim}$ results for each of the five methods. The second result is to compare the results based on the participation rate.

### 10.1. ATE Estimation for the Different Methods

For all the scenarios, five *ATE* quantities were compared with $ATE_{true}$: two baseline methods, Reg and Naive, and three X-Learner methods, LM, lasso, and RF. The results were compared using boxplots to visualize the distribution of predicted treatment effects.

Figure 3 presents results for where the ground truth treatment effect $ATE_{true}$ = 50, for both linear and non-linear simulations. The horizontal red lines in all the subsequent figures refer to $ATE_{true}$. In the linear data simulations, the traditional regression model, Reg, and the X-Learner linear model (LM) performed accurately whereas Naive, Lasso, and RF did not have any overlap in their distributions. Therefore, none of those mentioned had any estimations that equaled the true *ATE*. Naive appears to have performed the worst which is not surprising since it computed the difference between the mean treatment and control groups which contain bias.

For the non-linear simulations, again we find that Reg and LM perform the best. Lasso performed poorly. It showed a distribution that was even further from $ATE_{true}$ than for the linear results. RF on the other hand, appears to have more overlap with the true values, when compared with the RF linear results. This may be due being able to handle non-linearity well. The Naive results are now shown to have a much larger range of results with some overlap of the true values. This large range is most likely due to introducing non-linearity. For all non-linear results, there is a larger spread of values suggesting instability.

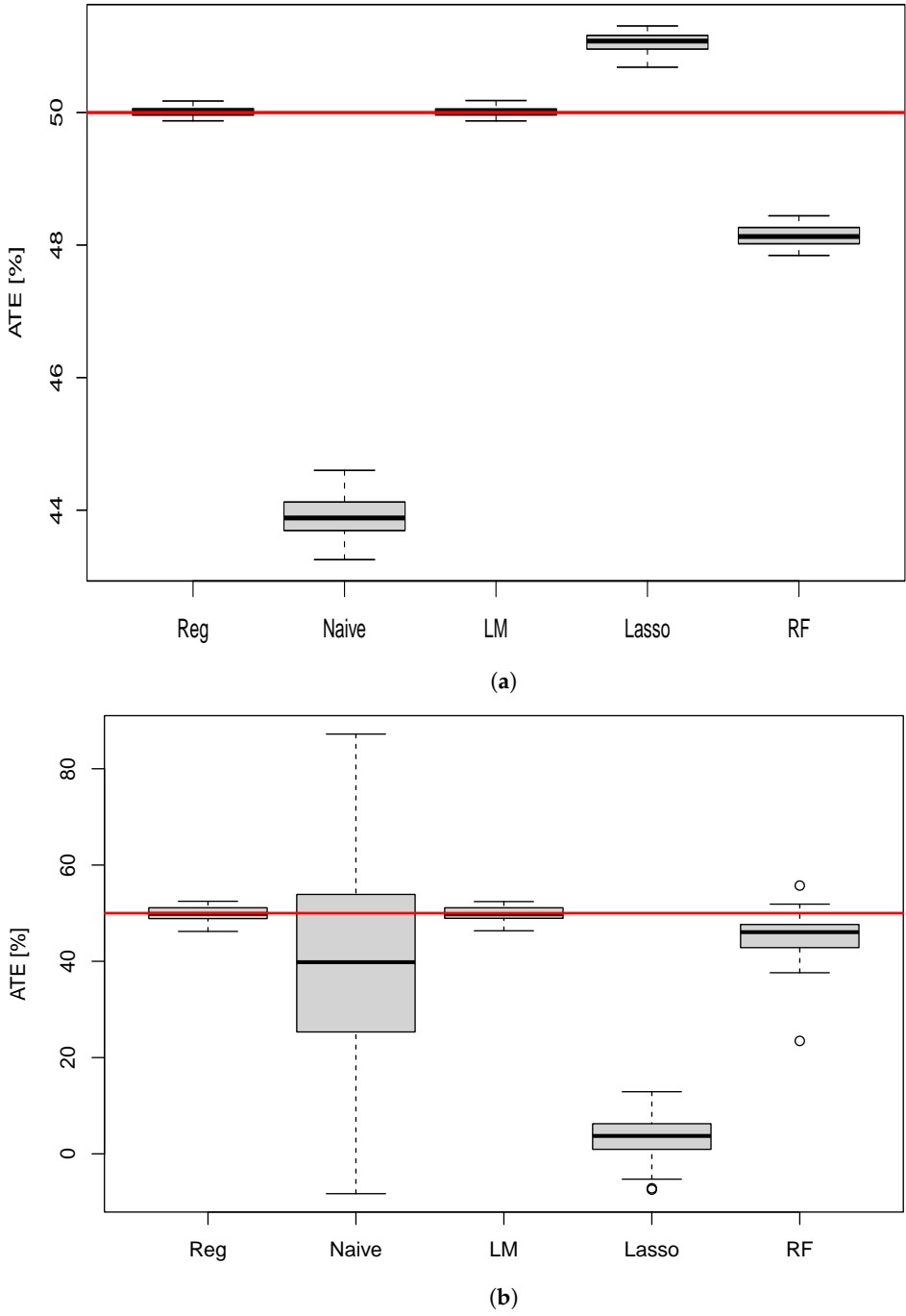

**Figure 3.** Box plots of the *ATE* estimations for the five methods where $ATE_{true}$ = 50. (**a**) linear. (**b**) non-linear.

For the remaining results for $ATE_{true}$ = 10 and 1 (Figures 4 and 5), we see similar patterns as those for $ATE_{true}$ = 50. Reg and LM performs well on both linear and non-linear data; Naive is the worst performer on the linear data but has overlap on the non-linear data; Lasso performs poorly on linear and non-linear (i.e., no overlap) and RF performs better on the non-linear data.

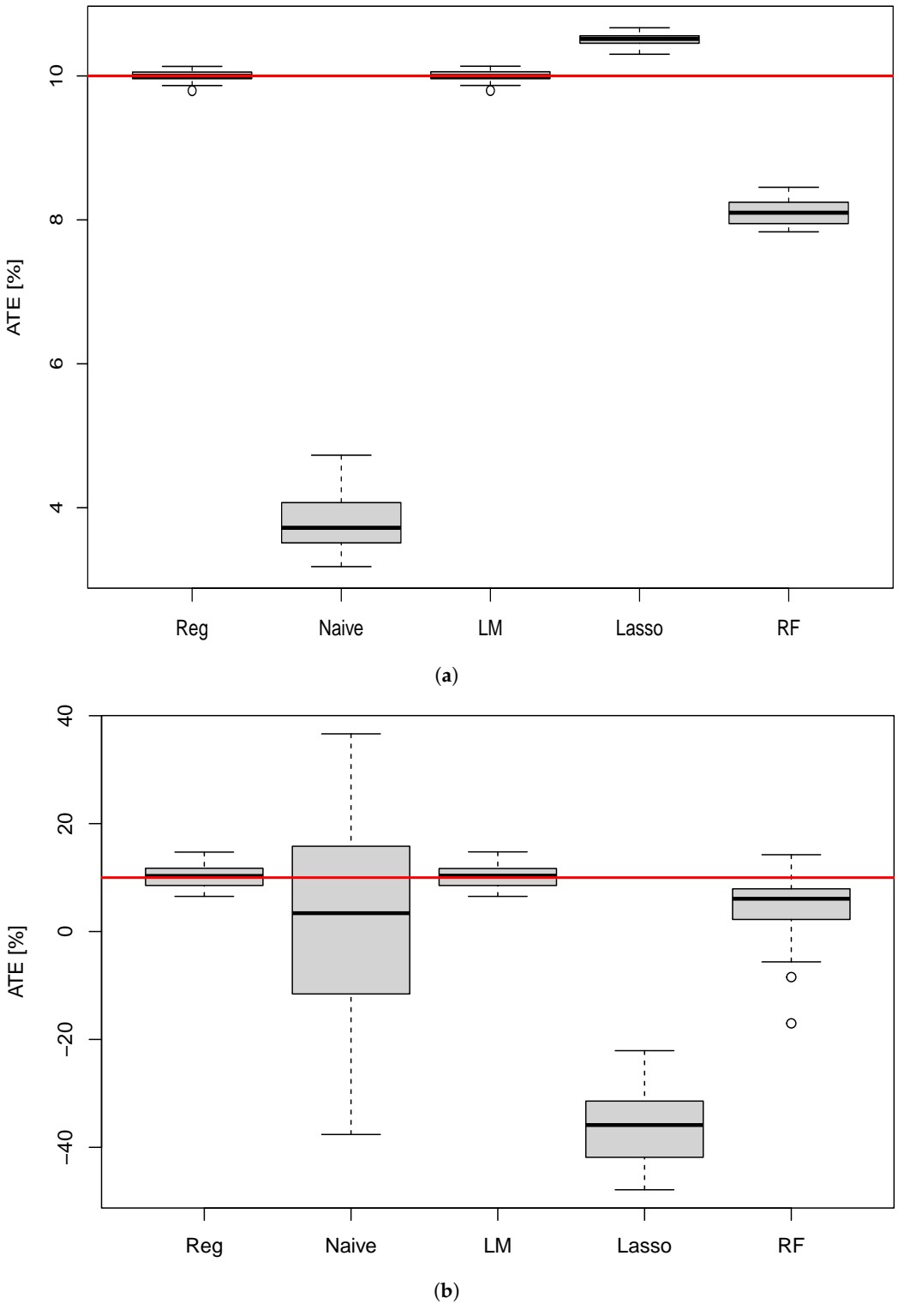

**Figure 4.** Box plots of the *ATE* estimations for the five methods where $ATE_{true}$ = 10. (**a**) linear. (**b**) non-linear.

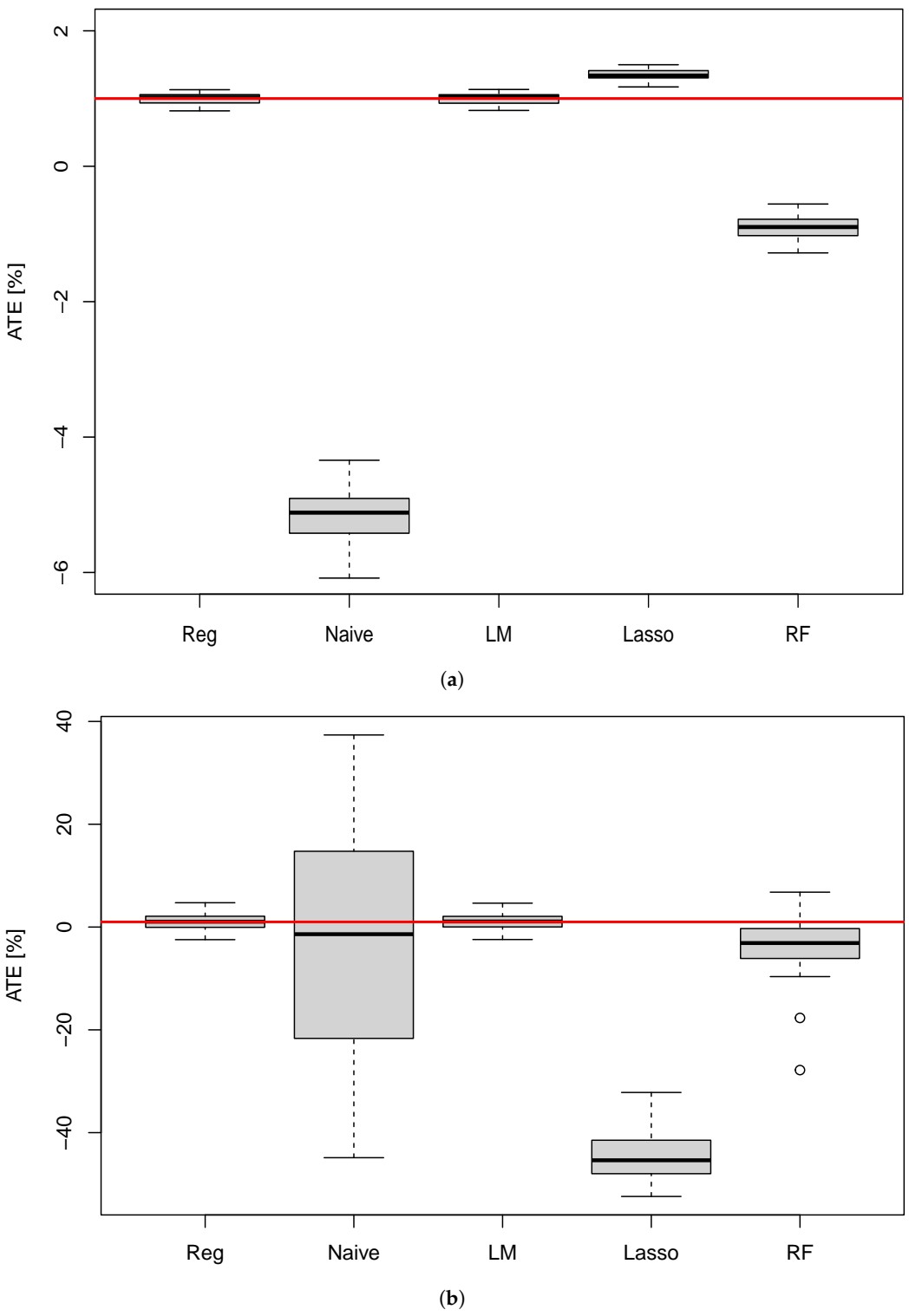

**Figure 5.** Box plots of the *ATE* estimations for the five methods where $ATE_{true}$ = 1. (**a**) linear. (**b**) non-linear.

Mean, standard deviation, and *t*-test *p*-value results for $ATE_{true}$ = 50, are presented in Tables 2 and 3. The results show that the *p*-values for Reg and LM are not significant, indicating they are not statistically different from $ATE_{true}$ (95% confidence interval). This means they predict the true treatment effect well. However, the rest of the methods were statistically different (*p*-value < 0.05).

**Table 2.** Summary results for $ATE_{true} = 50$, linear.

| Method | Mean | SD | *p*-Value |
|--------|------|-----|-----------|
| Naive | 43.0 | 19.9 | 0.00 |
| Reg | 50.0 | 0.06 | 1.28 |
| LM | 49.9 | 1.62 | 0.98 |
| Lasso | 51.0 | 0.11 | 0.00 |
| RF | 48.1 | 0.14 | 0.00 |

**Table 3.** Summary results for $ATE_{true} = 50$, non-linear.

| Method | Mean | SD | *p*-Value |
|--------|------|-----|-----------|
| Naive | 43.6 | 22.8 | 0.09 |
| Reg | 50.5 | 1.74 | 0.10 |
| LM | 50.0 | 1.66 | 0.10 |
| Lasso | 5.04 | 5.32 | 0.00 |
| RF | 44.1 | 0.14 | 0.00 |

*10.2. Effect of Participation Rate*

Here we look at how the participation rate affects the estimation of each method. For sake of brevity, we only look at $ATE_{true} = 10$. Figure 6 shows the results of the Naive method for participation rates of 10%, 50%, and 90%. For the linear dataset, the 50% rates show results closer to $ATE_{true}$ than for the more imbalanced participation rates. For non-linear, all three participation rates overlap $ATE_{true}$ but the 50% has the narrowest range and median line close to $ATE_{true}$. The results suggest that higher imbalance pushes results further from the true results and non-linearity introduces much larger ranges.

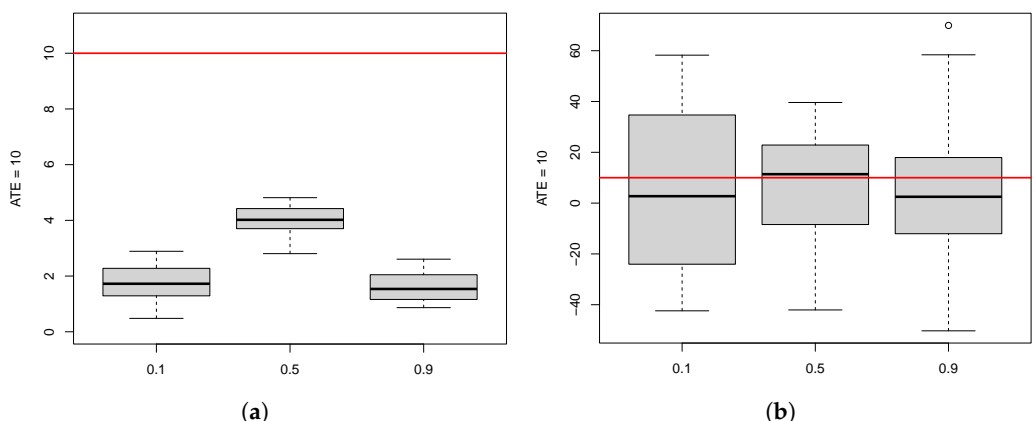

**Figure 6.** $ATE_{true} = 10$ estimations for different participation rates for Naive method. Note that the scales of the two figures are not the same. (**a**) Naive linear. (**b**) Naive non-linear.

For the traditional regression methods in Figure 7, the higher imbalance (0.1 and 0.9) result in larger ranges of *ATE* predictions; for both linear and non-linear. Therefore, degree of imbalance appears to affect the *ATE* prediction.

Moving on to the X-Learner LM method (Figure 8), higher imbalance (0.1 and 0.9) show larger ranges than 0.5, as before. This suggests that if there is large imbalance, this might cause overlap with the true value, but often it will be incorrect. The 0.5 rates have much narrower ranges. Again we see for non-linear data, the ranges are very large, compared with linear.

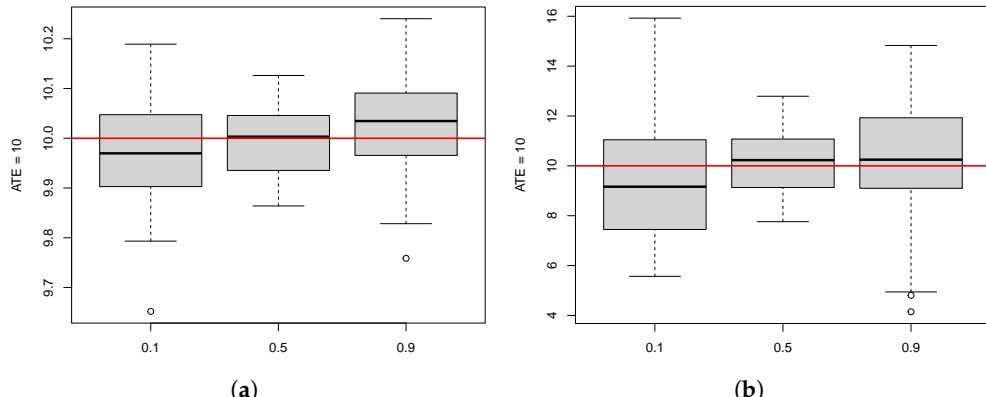

**Figure 7.** $ATE_{true}$ = 10 estimations for different participation rates for Regression method. Note that the scales of the two figures are not the same. (**a**) Reg linear. (**b**) Reg non-linear.

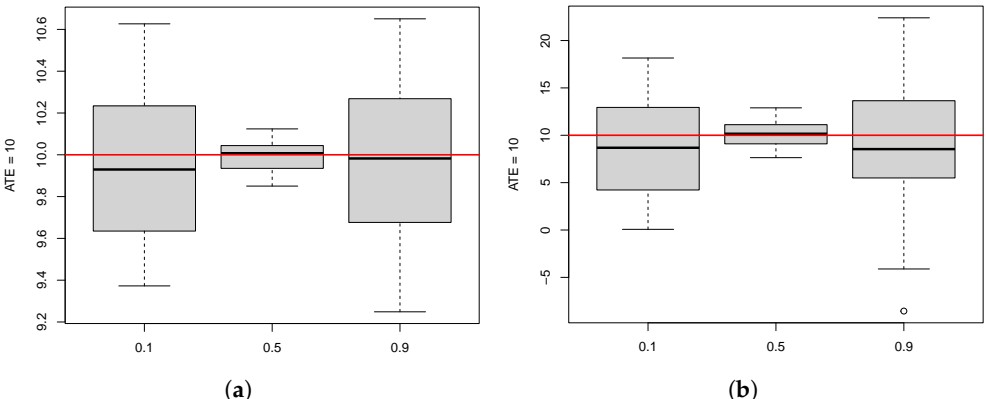

**Figure 8.** $ATE_{true}$ = 10 estimations for different participation rates for X-Learner LM method. Note that the scales of the two figures are not the same. (**a**) X-Learner LM: linear. (**b**) X-Learner LM: non-linear.

The results for X-Learner Lasso are shown in Figure 9. Higher imbalance is seen to push results further from $ATE_{true}$ and increases the spread of results. When we introduce non-linearity, higher imbalance is shown to push results egregiously far from $ATE_{true}$. In non-linearity, the error seems to be magnified considerably more.

Similar results are found for X-Learner RF, shown in Figure 10. Higher imbalance results in estimations that are further from the true value than the 0.5 and higher imbalance causes larger ranges of values.

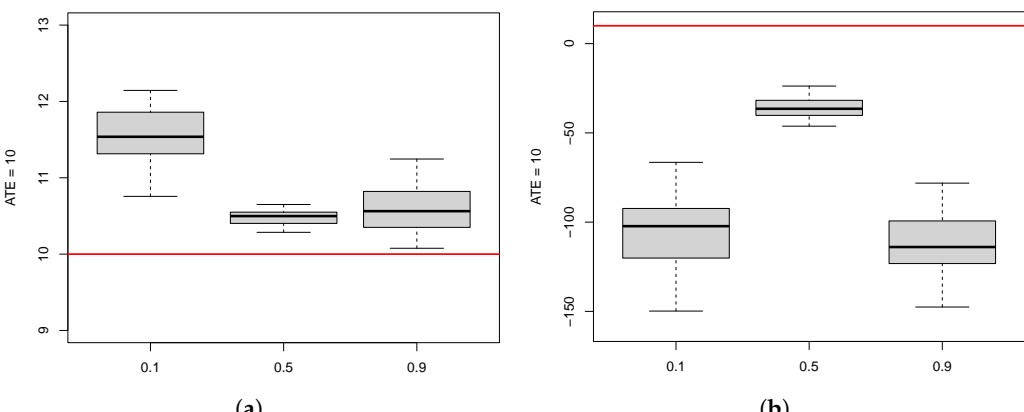

**Figure 9.** $ATE_{true}$ = 10 estimations for different participation rates for X-Learner Lasso method. Note that the scales of the two figures are not the same. (**a**) X-Learner Lasso: linear. (**b**) X-Learner Lasso: non-linear.

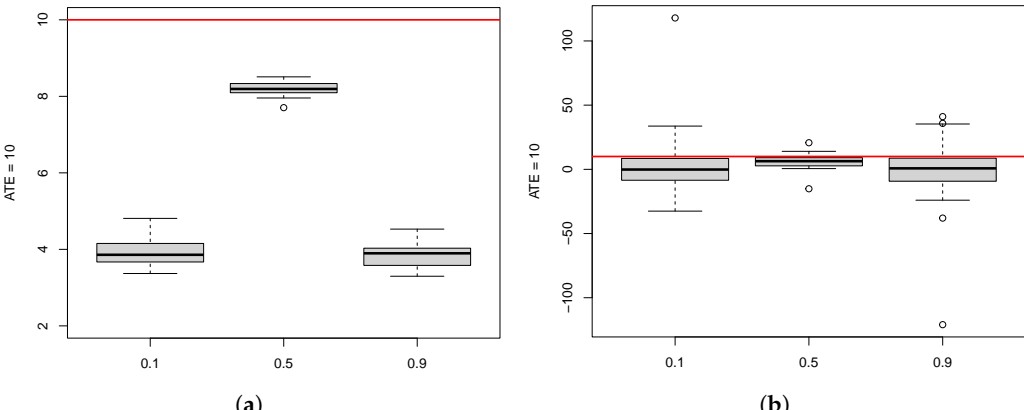

**Figure 10.** $ATE_{true}$ = 10 estimations for different participation rates for X-Learner RF method. Note that the scales of the two figures are not the same. (**a**) X-Learner RF: linear. (**b**) X-Learner RF: non-linear.

## 11. Discussion

This study presents, as far as we know, the first simulation study investigating the performance of the X-Learner method in the presence of observed confounding and non-linearity, as well as evaluating three supervised learning models and comparing it with traditional methods. It was vital to compare this method with a traditional method to determine if it is worth it to carry out the extra complex steps.

The main finding of this study is that the X-Learner presents no clear benefits over the traditional regression method, for the scenarios simulated here. It may outperform traditional methods under different scenarios not covered here, such as for non-constant treatment effects [11]. However, in our study, the traditional regression method of estimating treatment effects by computing the treatment indicator coefficient, outperforms the X-Learner method: in both linear and non-linear data simulations. Fitting a regression model and obtaining the treatment effect coefficient, provides sufficiently accurate results. The X-Learner LM (i.e., X-Learner with regression base learner) performs just as well as the traditional regression model; however, it requires extra work to compute and does not provide any clear benefits for that extra work. In their study where they introduce the X-Learner, Kunzel et al. [11] suggest that the X-Learner is able to perform well when the treatment effects are non-constant treatment effects *ATE* and when there is high imbalance. In our study, we kept our *ATE* constant so it is hard to compare our results with theirs. However, our results show that with higher imbalance, the X-Learner performs worse that with perfect balance.

As discussed earlier, the X-Learner Lasso and RF methods did not perform well at all. It was surprising that the X-Learner Lasso performed so poorly. Under linear conditions it was second best to the linear models. However, under non-linear conditions, its performance was egregious. The use of regularized regression, such as lasso or ridge regression, to adjust for confounding, has been used in the literature [41,42]. Franklin et al. [41] compared the use of ridge and lasso regression with high-dimensional propensity score estimations, to adjust for confounding in a simulation study. They showed that regularized regression methods generally performed worse than the traditional propensity score approaches. Lasso regression aims to eliminate multi-collinearity in models. So if two features are highly correlated, lasso would aim to remove one of them from the model. However, in causal inference, if a confounder exists in the data, then we would ideally like to include it in our model. Leaving a confounder out of the model will bias the treatment effect coefficient [43]. Therefore, it could be that because the treatment indicator *T* is a function of the confounder $x_3$ (see Equation (4)), there exists correlation between *T* and $x_3$. Therefore, when applying lasso as the base learner, it aims to reduce one of the features to zero and may therefore affect the estimation of the treatment effect. In our case there was a $-0.6$ correlation coefficient between *T* and $x_3$ and, therefore, might be the reason behind the

poor performance. Therefore, lasso may be fine if we had correlation existing between two features where one is not a confounder. However, not if one of the features is a confounder. This is why it is vital, in any causal inference study, to carry out some form of causal discovery where we can determine the true causal structure in the data. Furthermore, lasso regression is based upon the linear regression model, but uses a more restrictive method for estimating the coefficients (as discussed in Section 7). Lasso can, therefore, be less flexible than linear regression [36] and may contribute to why it does not outperform simple X-Learner LM.

Although not applying the same meta-learner method used in our study, [25] compared non-parametric Bayesian additive regression trees (BART) with traditional methods such as PSM and regression based methods, by fitting models to a treated group and control group separately. Whereas our method trains models on the treated and control groups and uses predicted counterfactuals as per Equation (2) [25], only trains models on the treated and control groups and uses Equation (1). Furthermore, the non-linear simulations in [25] meant the treatment response was non-linear (heterogeneous), whereas our study kept the treatment response constant (homogeneous) and made a single feature, $x_1$, non-linear. The treatment itself in [25] varied as the features varied. In the non-linear simulations in [25] linear regression performed poorly whereas the non-parametric BART method detects the non-linearity. This is in contrast to our method where linear regression outperforms our non-parametric RF method. This could be attributed to [25], making the treatment effect non-linear, as discussed above, and our work making a feature non-linear.

Although RF never outperforms regression and LM, it does perform better when non-linearity was introduced. This is most likely due its non-parametric nature that can handle non-linearity [36]. When training the random forest models, we aimed to tune the hyperparameters as finely as possible. However, tuning of these parameters is often a challenging task requiring implementing exhaustive grid search or random grid search methods with a wide range of random forest hyperparameters. This tuning problem may be a major contributing factor to the poor predictive performance. This is suggested for future work as an area for improvement.

To conclude, the results of this study suggest that within the framework we have constructed, traditional regression methods work better and are much simpler to apply to a dataset.

*Limitations and Recommendations for Future Work*

This section discusses the limitations of this study and how it presents opportunities for further work.

- The confounding simulation was based on a single observable confounding feature affecting both the treatment and output. We did not look at multiple confounding features. We anticipate that more confounding would introduce larger errors. Future work can look at more observable confounders.
- We assumed that the treatment effect was constant. This implies that all treated people experienced the same effect of the treatment, which is real life is unfeasible. Future work would introduce non-constant treatment effects into the data.
- We did not simulate hidden confounding which is common in observational studies. The assumption in this study was no hidden confounding. Future work could study the results of simulations that include hidden confounding by generating datasets that include the confounding variable but then exclude it when carrying out methods such as the X-Learner method.
- More research is needed to understand how different models, such as neural networks or boosting will perform. More thorough tuning of model hyperparameters is suggested for more complex models, such as neural networks, random forest, and boosting models.
- When estimating treatment effects using model predictions, we used the same model on both treated and control groups. For example, we used linear regression, or random

forest, on both groups and estimated treatment effects. Future work could look at mixing up the models. For example, using a lasso model on a treated group and a random forest on the control and then estimating treatment effects.

- In a more general sense, it is hoped that future work would incorporate causal inference into AutoML. AutoML refers to automating the training of machine learning models [44,45]. Currently, no literature was found that incorporates causal inference into AutoML applications and this serves as a promising future application. Microsoft has developed causal inference applications that promise to perform end-to-end causal inference from raw data.

## 12. Statements and Declarations

The authors have no relevant financial or non-financial interests to disclose. The authors have no conflicts of interest in this study.

**Author Contributions:** Conceptualization: B.I.S.; Methodology: B.I.S. and C.C.; Software: B.I.S. and C.C.; Validation: B.I.S. and J.H.B.; Formal analysis: B.I.S.; Investigation: B.I.S.; Resources: B.I.S. and J.H.B.; Writing—original draft preparation: B.I.S.; Writing—review and editing: B.I.S. and J.H.B.; Supervision: C.C. and J.H.B. All authors have read and agreed to the published version of the manuscript.

**Funding:** This research received no external funding.

**Data Availability Statement:** Data is unavailable for this project. However, the code used to generate the data can be found under https://github.com/BevanSmith/Causal-Inference (accessed on 18 January 2023).

**Conflicts of Interest:** The authors declare no conflicts of interest.

## Abbreviations

The following abbreviations are used in this manuscript:

| | |
|---|---|
| ATE | Average treatment effect |
| RF | Random forest |
| LM | Linear regression model |
| PSM | Propensity score matching |
| IPTW | Inverse probability of treatment weighting |
| BART | Bayesian additive regression trees |
| RCT | Randomized controlled trials |
| DAG | Directed acyclic graph |
| SCM | Structural causal model |

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
