# Peer review of "Treatment Effect Performance of the X-Learner in the Presence of Confounding and Non-Linearity"

_mca, doi:10.3390/mca28020032_

Round 1

Reviewer 1 Report

Major:
1. Section 5.1 - support the decision to the feature distribution from real-world data. For example, how do we know something is normally distribution or something else?

2. The authors encouraged to add recent online-learning studies that use ML to the introduction:
A. https://doi.org/10.1007/s43545-022-00337-4
B. https://doi.org/10.1080/09523987.2019.1669875 

3. "This is due to it being generally unethical to randomly assign students to receive (or not receive) an intervention." - These studies passed an ethical committee. Thus, it is a problematic and even wrong claim.  

4. I wanted to request the authors compared X-learner in their settings with AutoML as this is the latest development in the field from a performance point of view. However, this would result in a large amount of work which I am not sure is fair for this scope of work. Thus, I would just request the authors to discuss AutoML in their settings at the last section. To help, the authors can review and use, but not limited to, recent works:

Minor:
1. Abstract - "The main finding is that under the range of scenarios presented 8 in this study, the X-Learner at best matches the traditional regression methods and at worst performs 9 egregiously." - bad wording and not specific. I would be happy to get a clear cut to the cases it works well and not. 

2. Table 1 - X3 and X6 are both "grades" 

Compliments:
1. The abstract is well written.

2. The experiments are well designed and presented

Reviewer 2 Report

The article is very good and interesting. I recommend it for publication after minor revision. 

Comments and Suggestions for Authors

1. Introduction needs to explain the main contributions of the work more clearly. The difference between present work and previous. Works should be highlighted.

2. Comparison with recent study and methods would be appreciated What is the motivation of the proposed work. Also, a detailed literature review should been written.

3.There are many symbols and notations in paper. The author should explain these symbols and notations very well for the work to be understandable.

4. I would like to review some discussions about your future studies. For instance, the proposed methods can be extended to Fermatean fuzzy sets. 

5. There are many abbreviations in the work. Authors should include an abbreviation section.

Reviewer 3 Report

Please, see the attached file

Round 2

Reviewer 1 Report

The authors answered all my concerns. The paper can be published now as far as I consider 

Reviewer 3 Report

In my opinion, the paper, in the current revised version, is suitable for publication.